# Proximate analysis and vitamin B contents of fresh-made, canned chickpea and broad bean dips commercially produced in Jordan

**Seham M. Abu Jadayil**[1]*, **Ali K. Alsaed**[1], **Iman F. Mahmoud**[1,2], **Leena M. Ahmad**[1], **Fatena Afaneh**[1], **Hanaa Khalaf**[1], **Mohammed Z. Soudi**[3]

1 Department of Nutrition, Faculty of Pharmacy and Medical Sciences, University of Petra, Amman, Jordan,
2 Department of Basic Human Sciences, Faculty of Arts and Sciences, University of Petra, Amman, Jordan,
3 General Manager for Jordan Valley Food Industrial Company, Amman, Jordan

* sabujadayil@uop.edu.jo

## Abstract

**Data Availability Statement:** All relevant data are within the manuscript and its Supporting Information files.

### Background

Chickpea and broad bean dips are among the most popular legume-based dishes in the Middle Eastern countries. They are either made freshly by restaurants or sold in cans. Various manufacturing processes may enhance or reduce the chemical compositions of any food products, including these dips, which in turn can affect their nutritional values and health benefits. Therefore, this study aimed to evaluate the nutritional values of the chickpea and broad bean dips and examine the possible differences between those made freshly and those sold as canned products.

### Methods

Fresh-made and canned chickpea and broad bean dips were obtained from various restaurants and factories in Jordan and were analyzed for their proximate analysis, titratable acidity, and pH value. Furthermore, vitamins B were analyzed using liquid chromatography-mass spectrometry.

### Results

Significant differences were detected between fresh-made and canned chickpea dips, with the former containing higher quantities of fiber (10.96g ± 0.32) while the latter containing higher quantities of proteins (8.06g ± 0.29), fats (8.05g ± 1.08), and the vitamins B1 (0.46 ± 0.02) and B5 (0.87 ± 0.02). On the other hand, a significant difference was detected between fresh-made and canned broad bean dips, while the latter contained higher quantities of carbohydrates (20.94g ± 0.78) and tested B-vitamins (except for B6). These detected differences may be due to different variances of chickpeas and broad beans used, preparation methods, and/or the addition of other ingredients.

**Funding:** The Deanship of Research and Postgraduate Studies in Petra University. The funders had no role in study design, data collection and analysis, decision to publish, or preparation of the manuscript.

**Competing interests:** The authors have declared that no competing interests exist.

## Conclusion

Our results indicate that both chickpea and broad bean dips prepared/sold in Jordan were of high nutrition values in terms of proximate analysis, and vitamins B, with higher quantities observed in the canned dips. Higher titratable acidity and lower pH were also significantly found in the canned dips. This study adds to the existing literature regarding the fresh-made and canned chickpea and broad beans dips produced and sold in Jordan. Moreover, this study shows that canned chickpea and broad beans dips can provide consumers with comparable nutrient values to those provided by the freshly made dips. Nevertheless, these findings warrant more investigations.

## Introduction

Chickpea dips and broad bean dips are among the most popular legume-based dishes consumed by the Middle East countries such as Jordan, Syria, and Lebanon [1]. Chickpea dips, commonly known as *Hummus*, are made from dried chickpeas, tahini (sesame seed paste), lemon juice or citric acid, garlic, and salt [1, 2]. Broad bean dips, commonly known as *Ful Medames*, are dishes that are made from stewed broad beans seasoned with tahini, citric acid, and salt [1, 3]. These legume-based dishes are consumed for their delicious taste, low cost, and the high nutritional value of their ingredients [3, 4].

The main ingredient of chickpea dips and the broad bean dips are dried chickpeas and dried broad beans. Dried chickpeas (*Cicer arietinum* L.) from the Fabaceae family, are the main component used for the preparation of the chickpea dips. They are considered a rich source of proteins, carbohydrates, dietary fibers, vitamins (water soluble and insoluble vitamins), and minerals [1, 5]. On the other hand, dried broad beans (*Vicia faba* L.), also known as faba beans, horse beans, and field beans, are also a member of the Fabaceae family that has high concentrations of proteins and carbohydrates [6]. Moreover, they are significantly rich in minerals (e.g., zinc and iron), vitamins (e.g., tocopherol), and vital fatty acids (e.g., linolenic acid) [3, 7]. Indeed, broad beans are regarded as high-density food items rich in vital nutrients as well as low-calorie contents.

The rich chemical compositions and nutritive values of chickpeas and broad beans made these legumes an essential part of the daily diet of many low and middle-income countries [1, 3]. Moreover, chickpeas and broad beans have been reported to exhibit numerous health benefits attributed to their vital nutritive and non-nutritive components [5]. Aside from being an inexpensive source of vitamins, minerals, and other bioactive components, chickpeas have been reported to possess potential preventive activities against cardiovascular diseases, diabetes mellitus, cancer, and obesity, among other chronic diseases [4, 8]. Likewise, literature has documented the various health-promoting properties of broad beans. Extracts of broad beans have shown antioxidant, antimicrobial, anti-diabetic, anticancer, and hypoglycemic activities [9, 10].

Chickpea dips and broad bean dips have been reported to contain the same important nutritive and vital bioactive components, hence the health benefits similar to those exhibited by chickpeas and broad beans [2, 3]. Nevertheless, it is important to note that various processing methods may enhance or reduce the chemical compositions of chickpeas and broad beans, and hence affect their nutritional values as well as their health benefits [11, 12]. Moreover, chickpea dips and broad bean dips commercially sold in the market do not usually follow the

same traditional recipe and/ or added ingredients as those freshly made [13, 14]. It is well-known that the composition of commercially sold chickpea dips and broad bean dips is continuously being modified to produce healthier and tastier products as well as to increase their self-lives [14]. On one hand, these modifications (e.g., the addition of nutrients and chemicals) may enhance the nutritional values of these dips. Furthermore, certain processing modifications may result in the negative alteration of these legumes-based dips [13].

Understanding the importance of these legume-based dips as an essential part of the traditional diet consumed in Jordan and the undeniable effects of processing methods on these dips. This study was carried out to determine the proximate analysis and vitamin B contents of different types of fresh-made and canned chickpea and broad bean dips that are commercially produced and sold in Jordan.

## Materials and methods

### Selection and preparation of chickpea and broad bean dips samples

Six batches of commercially plain chickpea and broad bean dips were obtained from 3 restaurants and 3 factories that specialized in manufacturing these commercial foods in Amman, Jordan. We communicated by email or phone with the most commercial factories and restaurants found in Jordan (7 factories, 7 restaurants). The research team then visited several processing factories and restaurants producing chickpea and broad bean dips, as well as a several hospitals and hotels. Only three processing factories and three restaurants participated in the study after explaining the study objectives, while the rest declined participation. Triplicate samples of fresh-made chickpeas and broad bean dips were collected from each restaurant, whereas triplicates of canned chickpeas and broad beans were obtained from each factory. The triplicate samples of the selected fresh/ canned chickpeas/broad bean dips were mixed well to get a representative sample for each group and were directly processed to produce powder forms.

### Preparation of chickpea and broad bean dip powder

The representative samples of chickpea and broad bean dips were mixed correctly to obtain homogenized samples. These samples were then dried using a vacuum drying oven at 100˚C / 3 hours (vacuum drying oven, RAYPA model: EV-50, Barcelona, Spain), and then milled using a grain mill with 0.5-mm sieve into powder forms. The resultant powders were then kept in polyethene bags and stored in a deep freezer at -18˚C until further analysis. Triplicate samples were used to determine the chemical composition.

### Proximate composition analyses

The following analyses were done according to the official method (AOAC, 2000) [15] and the ASEAN Manual of Food Analysis (2011) [16].

**Moisture.** In brief, 5 grams of tested samples were placed on the pre-dehydrated aluminum dish. Samples were then dried in a vacuum oven at 100˚C for 3 hours. Samples were then cooled in the desiccator and weighed. The moisture of the samples was calculated based on the following equation:

$$Moisture \; (grams)\% = (W2 - W3 / W2 - W1) \; x \; 100\%$$

Where:
W1: weight of the empty pre-dehydrated aluminum dish
W2: weight of the pre-dehydrated aluminum dish and samples before dryness

W3: weight of the pre-dehydrated aluminum dish and samples after dryness

**pH.** The sample's pH was determined using a digital pH meter (inoLab pH 7110 Waterproof Pocket Tester, Germany) and calibrated with pH 4.0 and 7.0 buffers.

**Crude protein.** Samples of crude protein were determined using the Kjeldahl method. In brief, 2 grams of each sample was digested by mixing it with $H_2SO_4$ solution and a catalyst tablet (potassium sulfate and copper sulfate) and heated at 370–400°C for 3 hours until a clear solution was achieved. The distillation step was then achieved by adding 20 ml of distilled water in a 50 ml of boric acid (4%) with a phenol phthalic indicator as a receiver on the distillation unit, followed by the addition of NaOH (50%). Finally, the solution was titrated with 0.1 N HCl until a pink color appeared and the volume of HCl used was recorded. The crude protein of the samples is calculated based on the following equations:

$$N(\%) = \frac{(Volume\ of\ HCl\ sample - volume\ of\ HCl\ blank)\ x\ 14.007\ x\ 0.1N\ HCl}{1000\ x\ Weight\ of\ sample} \tag{1}$$

$$\% \ protein = nitrogen\% \ x \ 6.25 \tag{2}$$

Where:

N: nitrogen

**Crude fat.** In brief, 1–2 grams of dried sample were placed onto a dried thimble. Samples were then mixed with diethyl ether solution, and heated to 70°C for 1 hour, followed by an additional 1 hour at 120°C. Afterward, the samples were subjected to dryness in a vacuum oven for 10 minutes, left to cool and the weight of the extracted fat was recorded. The sample's crude fat was expressed using the following equation:

$$Crude\ fat\ (grams) = (W2 - W1/W)\ x\ 100$$

Where:

W: weight of the dried sample

W1: weight of the empty dried beaker without the thimble

W2: weight of the dried beaker with the thimble and sample after extraction

**Crude fiber.** In brief, 1 gram of dried sample was placed onto a dried crucible. Samples were mixed with $H_2SO_4$ (1.25%) and boiled three times, each time for 30 minutes. Then, NaOH (1.25%) was added to the mixture and was boiled for an additional 30 minutes/3 times. Samples were then washed with absolute acetone, and then subjected to dryness in the oven and weight was recorded. The crucible was then ashed in a muffle furnace at 550°C and weighed. The sample's crude fiber was expressed using the following equation:

$$Crude\ fiber\ (grams) = (F1 - F2/F0)\ x\ 100$$

Where:

F0: weight of the dried samples and dried crucible

F1: weight of the dried samples and dried crucible after dryness in the oven

F2: weight of the dried samples and dried crucible after ashing by muffle furnace

**Ash.** In brief, the dried crucible was weighed and 3–5 grams of samples were added to the crucible and weighed again. Samples were then ashed by the muffle furnace at 500–550°C until a white ash was obtained. White ash is then weighed again. Sample ash was calculated using the following equation:

$$Ash\ (grams) = (W3 - W1/W2 - W1)\ x\ 100$$

Where:

W1: weight of the dried crucible
W2: weight of the dried samples and dried crucible before muffle furnace
W3: weight of the dried samples and dried crucible after muffle furnace

**Carbohydrate.** Samples' carbohydrate content was calculated based on the following equation:

$$Carbohydrate\ (grams) = [100 - (moisture + fat + protein + ash + crude\ fiber)]$$

**Titratable acidity.** In brief, 6 grams of samples were diluted in 50 ml of distilled water and titrated by NaOH (0.1N) to a pH of 8.2. The titratable acidity was then expressed as gram citric acid/kg sample using the following equation:

$$Titratable\ acidity\ (g\ citric\ acid/kg\ of\ the\ sample)$$
$$= (V \times 0.1N \times 1000 \times 0.064\ eq.wt.of\ citric\ acid)/weight\ of\ the\ sample$$

Where:
g: gram
N: normality
kg: kilograms
V: volume
eq. wt.: equivalent weight

## Vitamin B contents

The determination of the vitamin B contents in the samples was conducted using liquid chromatography-mass spectrometry (LC-MS). In brief, the samples were homogenized, and 10 grams were placed in 50 ml centrifuge tubes. To these samples, 2–20 mL ultrapure water was added, and immediately covered with aluminum foil to protect the samples from light. Samples were then shaken for 10 minutes and then subjected to ultrasonication for 15 minutes. Samples were then centrifuged for 5 minutes (5000 rpm) and the resultant supernatants were filtered through 0.45 um nylon filter. Around 3 µl of the filtrate was injected into LC-MS/MS for analysis. A Bruker Daltonik (Bremen, Germany) Impact II ESI-Q-TOF System equipped with Bruker Dalotonik (Bremen, Germany) was used for screening the compounds, using direct injection. Standards were used with high-resolution Bruker TOF MS and the exact of each analyte. This instrument was operated using the Ion Source Apollo II ion Funnel electrospray source. The capillary voltage was 2500 V, the nebulizer gas was 2.0 bar, the dry gas (nitrogen) flow was 8 L/min and the dry temperature was 200˚C. The mass accuracy was <1 ppm; the mass resolution was 50000 FSR (Full Sensitivity Resolution) and the TOF repetition rate was up to 20 kHz. The procedure was done by Naba Hikma For Industrial and Testing Services, Amman, Jordan.

## Statistical analysis

The statistical analysis was performed using the Statistical Analysis System (SAS) version 9.0. Differences were significant at $P < 0.05$. Results were expressed as mean ± standard error of the mean (SEM). One-way ANOVA was used to detect the differences in the means between the fresh-made samples and the canned chickpea/ broad bean dips, and the differences of means between both dips (chickpea and broad bean), followed by Fisher's Protected LSD mean separation test [17].

**Table 1. Proximate analysis (g /100 g wet matter), titratable acidity and pH of fresh-made and canned chickpea dip samples.**

| Variable | Fresh-Made Chickpea Dips | | | P- value | Canned Chickpea Dips | | | P- value |
|---|---|---|---|---|---|---|---|---|
| | Sample 1 | Sample 2 | Sample 3 | | Sample 1 | Sample 2 | Sample 3 | |
| | Mean±SEM | Mean±SEM | Mean±SEM | | Mean±SEM | Mean±SEM | Mean±SEM | |
| Ash wet (g) | 1.59±0.15 | 2.32±0.15 | 3.58±0.15 | 0.0002 | 1.67±0.02 | 1.59±0.02 | 1.59±0.02 | 0.0415 |
| CHO (g) | 14.17±1.27 | 26.60±1.03 | 11.99±1.03 | 0.0004 | 16.79±0.31 | 15.19±0.31 | 19.05±0.26 | 0.0016 |
| Crude fiber (g) | 10.00±0.21 | 8.93±0.21 | 10.00±0.21 | 0.0154 | 12.50±0.21 | 10.00±0.21 | 10.37±0.21 | 0.0003 |
| Moisture (g) | 66.00±0.58 | 58.37±0.59 | 66.47±0.59 | 0.0001 | 70.00±0.39 | 71.33±0.39 | 70.00±0.39 | 0.0787 |
| Protein (g) | 8.03±0.19 | 9.33±0.19 | 6.80±0.19 | 0.0002 | 7.10±0.09 | 6.43±0.09 | 6.13±0.09 | 0.0008 |
| Crude Fat wet (g) | 10.37±0.85 | 3.39±0.70 | 11.17±0.70 | 0.0011 | 4.50±0.14 | 5.80±0.14 | 3.23±0.12 | 0.0004 |
| Titratable Acidity (%) | 0.53±0.03 | 0.62±0.03 | 0.50±0.03 | 0.0531 | 0.39±0.01 | 0.69±0.00 | 0.50±0.01 | <0.0001 |
| pH | 4.57±0.02 | 4.80±0.02 | 4.80±0.02 | 0.0002 | 4.53±0.03 | 4.23±0.03 | 4.50±0.03 | 0.0004 |

Results are expressed as mean of three replicates ± SEM.

SEM: standard error of mean; CHO: carbohydrates; g: grams.

## Results

### Proximate composition analyses, acidity, and pH of fresh-made and canned chickpeas samples

The proximate analyses, acidity, and pH of the 3 fresh-made samples and 3 canned samples of chickpea dips are shown in Table 1. For the fresh-made chickpea dips' samples, all proximate analyses (ash, CHO, fiber, moisture, protein, and fat) as well as the pH values were significantly different between the fresh-made samples tested except for the titratable acidity ($p = 0.0531$). Likewise, canned samples of chickpea dips showed significant differences in ash, CHO, fiber, protein, fat, and pH ($p < 0.05$). In contrast to fresh-made chickpea dip samples, canned chickpea dip samples were significantly different in the titratable acidity ($p < 0.001$) but showed insignificant differences in the moisture contents ($p = 0.0787$). For more information, please see Table 1.

### Comparison between fresh-made and canned chickpea samples

Proximate analyses, acidity, and pH comparison between fresh-made and canned chickpea dip samples are shown in Table 2. No significant differences in the titratable acidity and

**Table 2. Comparison of proximate analysis, titratable acidity, and pH between fresh-made and canned chickpea dips.**

| Variable | Fresh-Made Chickpea Dips | Canned Chickpea Dips | P-value |
|---|---|---|---|
| | Mean±SEM | Mean±SEM | |
| Ash wet(g) | 1.62±0.21 | 2.49±0.21 | 0.0098 |
| CHO (g) | 17.30±2.08 | 18.01±1.95 | 0.8067 |
| Crude fiber (g) | 10.96±0.32 | 9.64±0.32 | 0.0106 |
| Moisture (%) | 70.44±0.97 | 63.61±0.97 | 0.0001 |
| Protein (g) | 6.56±0.29 | 8.06±0.29 | 0.0020 |
| Crude Fat wet (g) | 4.33±1.15 | 8.05±1.08 | 0.0344 |
| Titratable Acidity (%) | 0.53±0.04 | 0.55±0.04 | 0.6088 |
| pH | 4.42±0.05 | 4.72±0.05 | 0.0002 |

Results are expressed as Mean of three replicates ± SEM.

SEM: standard error of mean; CHO: carbohydrate.

Table 3. Proximate analysis (g /100 g wet matter), titratable acidity, and pH of fresh-made and canned broad bean dip samples.

| Variable | Fresh-Made Broad Bean Dips | | | P- value | Canned Broad Bean Dips | | | P- value |
|---|---|---|---|---|---|---|---|---|
| | Sample 1 | Sample 2 | Sample 3 | | Sample 1 | Sample 2 | Sample 3 | |
| | Mean±SEM | Mean±SEM | Mean±SEM | | Mean±SEM | Mean±SEM | Mean±SEM | |
| Ash wet(g) | 1.40±0.05 | 0.81±0.05 | 3.37±0.05 | <0.0001 | 1.64±0.04 | 1.7±0.04 | 1.80±0.04 | 0.0477 |
| CHO (g) | 17.10±0.65 | 16.53±0.53 | 14.86±0.53 | 0.0827 | 21.20±0.51 | 23.58±0.51 | 18.03±0.51 | 0.0105 |
| Crude fiber (g) | 20.00±0.54 | 16.33±0.54 | 18.17±0.54 | 0.0086 | 20.00±0.17 | 23.57±0.17 | 12.60±0.17 | <0.0001 |
| Moisture (%) | 72.00±0.52 | 75.83±0.52 | 75.50±0.52 | 0.0036 | 67.90±0.39 | 65.93±0.39 | 72.67±0.39 | <0.0001 |
| Protein (g) | 8.03±0.19 | 6.43±0.19 | 4.30±0.19 | <0.0001 | 7.80±0.07 | 7.07±0.07 | 6.93±0.07 | 0.0003 |
| Crude Fat wet (g) | 1.40±0.40 | 0.40±0.33 | 1.97±0.33 | 0.0486 | 1.61±0.01 | 1.70±0.01 | 0.27±0.01 | <0.0001 |
| Titratable Acidity (%) | 0.18±0.01 | 0.10±0.01 | 0.16±0.01 | 0.0066 | 0.10±0.001 | 0.08±0.001 | 0.09±0.001 | <0.0001 |
| pH | 6.07±0.02 | 5.97±0.02 | 6.18±0.02 | 0.0014 | 5.70±0 | 5.70±0 | 6.10±0 | <0.0001 |

Results are expressed as Mean of three replicates ± SEM.

SEM: standard error of mean; g: grams; CHO: carbohydrate.

carbohydrate content were detected between samples of fresh-made and canned chickpea dips ($p > 0.05$). On the other hand, there was a significant difference in ash, crude fiber, protein, crude fat contents, and pH value between samples of canned and fresh-made chickpea dips ($p < 0.05$). The mean value of crude fiber (10.96g ± 0.32) and moisture contents (70.44g ± 0.97) of fresh-made samples of chickpea dips was higher than that found in canned samples (9.64g ± 0.32 and 63.61g ± 0.97, respectively). Meanwhile, the mean value of ash (2.49g ± 0.21), protein (8.06g ± 0.29), and fat contents (8.05g ±1.08) of canned samples of chickpea dips were higher than those detected in the fresh-made samples (1.62g ± 0.21, 6.56g ±0.29, and 4.33g ± 1.15, respectively). As for the pH values, canned samples of chickpea dips (4.72 ± 0.05) scored higher values than those found in fresh-made samples (4.42 ±0.05). For more information, please see Table 2.

## Proximate composition analyses, acidity, and pH of broad bean samples

The proximate analysis, titratable acidity, and pH of fresh-made and canned broad bean dip samples are shown in Table 3. For the fresh-made broad bean dips' samples, all proximate analyses (ash, fiber, moisture, protein, and fat) as well as the titratable acidity and pH value were significantly different between the 3 tested fresh-made samples except for the CHO content ($p = 0.0827$). Canned samples of broad bean dips, on the other hand, showed significant differences in all proximate analyses, titratable acidity, and pH value among the 3 tested samples ($p < 0.05$). For more information, please see Table 3.

## Comparison between fresh-made and canned broad bean samples

The comparison of proximate analysis, titratable acidity, and pH between canned and fresh-made broad bean dips is shown in Table 4. No significant differences in ash, crude fiber, protein, as well as fat contents between tested samples of fresh-made and canned broad bean dips ($p > 0.05$). Whereas a significant difference between tested samples of fresh-made and canned broad bean dips was detected in the carbohydrate and moisture contents, as well as the titratable acidity and pH value ($p < 0.05$). The titratable acidity, moisture content, and pH value were higher in fresh samples (0.15 ± 0.01, 74.44g ± 0.68, and 6.07 ± 0.05, respectively) than in the canned samples (0.09 ± 0.01, 68.83g ± 0.68, and 5.83 ± 0.05, respectively) of broad bean dips. On the other hand, the carbohydrate content of canned samples (20.94g ± 0.78) was

**Table 4. Comparison of proximate analysis, titratable acidity, and pH between canned and fresh-made broad bean dips.**

| Variable | Fresh-Made Broad Bean Dips | Canned Broad Bean Dips | P-value |
|---|---|---|---|
| | Mean±SEM | Mean±SEM | |
| Ash wet(g) | 1.86±0.28 | 1.72±0.28 | 0.7251 |
| CHO (g) | 16.05±0.68 | 20.94±0.78 | 0.0005 |
| Crude fiber (g) | 18.17±1.65 | 18.72±1.35 | 0.7981 |
| Moisture (%) | 74.44±0.86 | 68.83±0.86 | 0.0003 |
| Protein (g) | 6.26±0.40 | 7.27±0.40 | 0.0932 |
| Crude Fat wet (g) | 1.24±0.29 | 1.19±0.33 | 0.9169 |
| Titratable Acidity (%) | 0.15±0.01 | 0.09±0.01 | 0.0005 |
| pH | 6.07±0.05 | 5.83±0.05 | 0.0052 |

Results are expressed as Mean of three replicates ± SEM.

SEM: standard error of mean; g: grams; CHO: carbohydrate.

higher than that of the freshly made broad bean dip samples (16.05 g ± 0.68). For more information, please see Table 4.

## Vitamin B contents

The B vitamin contents of the tested samples of fresh-made and canned chickpea and broad bean dips are shown in Table 5. Significant differences in vitamin B contents (nicotine, pantothenic acid, pyridoxine, riboflavin, and thiamin) were detected between fresh-made and canned samples of chickpea ($p < 0.001$). Likewise, a significant difference in vitamin B contents was detected between fresh-made and canned samples of broad bean dips except for pyridoxine vitamin ($p = 0.8372$). In general, fresh-made samples of chickpea and broad bean dips had higher vitamin B contents than the canned ones. The highest difference in vitamin B contents between the fresh-made and canned samples of both chickpea and broad bean dip samples was detected for the thiamin vitamin (0.4130 and 0.5626mg, respectively). For more information, please see Table 5.

## Discussion

Chickpea (Hummus) and broad bean (Ful Medames) dips are traditional and nutritious foods that are popular in the Middle East, with growing popularity globally [1, 7]. To accommodate

**Table 5. Vitamin B contents (mg /100 g wet matter) of fresh-made and canned chickpea and broad bean dip samples.**

| Vitamin B Contents | Chickpea Dips | | P- value | Broad bean Dips | | P- value |
|---|---|---|---|---|---|---|
| | Fresh-Made | Canned | | Fresh-Made | Canned | |
| | Mean±SEM | Mean±SEM | | Mean±SEM | Mean±SEM | |
| Thiamin (B1, mg) | 0.04227±0.01862 | 0.4553±0.01862 | <0.0001 | 0.02930±0.01862 | 0.5919±0.01862 | <0.0001 |
| Riboflavin (B2, mg) | 0.1218±0.003099 | 0.1474±0.003099 | 0.0004 | 0.06517±0.003099 | 0.1064±0.003099 | <0.0001 |
| Niacin (B3, mg) | 0.2120±0.02092 | 0.6927±0.02092 | <0.0001 | 1.9650±0.02092 | 3.8741±0.02092 | <0.0001 |
| Pantothenic acid (B5, mg) | 0.5064±0.01863 | 0.8699±0.01863 | <0.0001 | 0.9739±0.01863 | 0.5488±0.01863 | <0.0001 |
| Pyridoxine (B6, mg) | 0.01492±0.000844 | 0.03988±0.000844 | <0.0001 | 0.03668±0.000844 | 0.03642±0.000844 | 0.8372 |

Results are expressed as Mean of three replicates ± SEM.

SEM: standard error of mean; mg: milligrams.

the high demand for these ready-to-eat foods, restaurants prepare large quantities of these dips daily, which are manufactured into cans by factories [12, 18, 19]. Various preparation and processing methods, alongside the different genotypes of chickpeas and broad beans, have been reported to significantly affect the chemical compositions and nutritional values of these legumes' dips [11, 12]. Yet limited studies have been done, comparing freshly made and canned chickpea and broad bean dips. Hence, the primary aim of this study was to determine the proximate analysis, titratable acidity, and pH values of several types (fresh and canned) of chickpea and broad bean dips, that are commercially produced and sold in Jordan. In addition, the vitamin B contents of these products were compared. The practical aim of this study was to examine the effect of the canning processes on both products' proximate analysis and vitamin B contents.

With regards to the proximate analysis of the chickpea dips, Alvarez *et al.* (2017) reported that commercially sold Spanish chickpea dips had a high fiber content of 8.40g/100g [14], whereas Wallace *et al.* (2016) reported that the fiber content of commercially chickpea dips sold in the United States was around 6.00g/100g [13]. The fiber content of both the commercial fresh-made and canned chickpea dips in this study was higher reaching up to almost 13.00g/100g. The difference in the fiber content detected could be attributed to the different chickpea varieties and processing methods used in the preparation of chickpea dips [2, 20]. Summo *et al.* (2019) reported a wide range of fiber content (11.00–22.10g/100g) was detected in the global chickpea collection tested. This was attributed to the chickpeas' variant (genotype and phenotype traits), which in turn resulted in differences in their chemical compositions [21]. Moreover, studies have revealed that preparation methods namely soaking and cooking; commonly used in preparing chickpea dips; increased significantly the overall fiber contents in these dips [22, 23]. It has been suggested that soaking causes partial solubilization of insoluble dietary fibers in chickpeas coupled with an increase in soluble dietary fibers. Whereas cooking increases insoluble dietary fibers due to the formation of resistant starch and/ or occurrence of Maillard's reaction [22, 23]. These findings may explain the high fiber content found in this study.

A study by Al-Awwad *et al.* (2014) detected high contents of proteins (6.24g/100g), carbohydrates (13.61g/100g), and fat (10.80g/100g) in the tested Jordanian chickpea dips [24]. Makhloufi and Yamani (2024) also reported high contents of proteins (7.46%), carbohydrates (14.48%), and fat (9.41%) in the freshly prepared chickpea dip [25]. Our results showed similar protein contents detected in both fresh-made and canned chickpea dips. In contrast, the carbohydrate contents of the fresh-made and canned chickpea dips were higher than those reported in these studies. Moreover, lower contents of fats were found, especially in the fresh-made chickpea dips. It is noteworthy that legumes such as chickpeas are low in fat and free of cholesterol [25]. Thus, the main source of fat in these legumes' dips is attributed to the second major dip component, sesame seed paste (tahini) [25, 26]. The addition of sesame seed paste at different percentages may have resulted in the differences in fat contents found in the tested chickpea dips in our study and the previous studies [26].

The popularity of canned food including canned chickpea dips has increased significantly among consumers globally [19, 27, 28]. Convenience, safety, and extended shelf-life are among the major advantages of canned products. However, canned products are also susceptible to quality deterioration due to bleaching and thermal treatments as well as the addition of chemical preservatives, which lead to loss of nutrients and flavors [27, 28]. On the other hand, studies have also revealed that soaking and thermal treatments done in the canning process break down the matrix of the food and reduce the antinutritional factors, increasing the release of major nutrients such as proteins and fats [27, 29]. Canned industries also tend to add natural and chemical components (i.e., citric acid, lemon juice) that increase the titratable acidity and

reduce the pH of the canned products. These additions enhance flavor and improve the perseveration capacity of the canned products [29]. In our study, significant differences in crude fiber, protein, and fat contents ($p < 0.05$) were found between fresh-made chickpea dips and canned chickpea dips, with higher protein and fat contents detected in canned chickpea dips and higher crude fiber content in fresh-made chickpea dips. Moreover, a significant difference ($p < 0.001$) in pH values between fresh-made chickpea dips and canned chickpea dips with the former having a lower pH. This may be due to the possible addition of higher amounts of lemon juice/citric acid to the freshly prepared dips. To reduce microbial growth, canned products tend to have reduced water content [30]. This was evident by the significantly lower moisture content detected in canned chickpea dips compared to that in the fresh-made dips.

With regards to broad bean dips, Al-Faris (2017) reported that broad bean dips found in Saudi Arabia had 7.96% protein, 17.15% carbohydrate, 3.20% fat content, and 0.50% fiber content [31]. Another study by Kirse and Karklina (2015) showed that broad bean spreads tested were considered good sources of proteins (9.21–9.59g/100g) and crude fibers (14.23g/100g) [32]. According to the USDA Food Data [33] central database, broad bean dips contain 4.5g/100g fibers, 6.82g/100 proteins, 12.88g/100g carbohydrates, and 0.38g/100 fats. Our results showed similar protein contents detected in both fresh-made and canned broad bean dips. However, the fiber, carbohydrate, and fat contents of the fresh-made and canned broad bean dips were higher than those reported previously. This may be explained by the different preparation methods used and the variance of the broad beans used in these dips [3, 34]. The only significant difference ($p < 0.001$) detected between the fresh-made and canned broad bean dips was in the carbohydrate content. This difference may be attributed to the dry matter of the dips, in which the higher the moisture content, the lower the amount of carbohydrate content, and vice versa [2]. Indeed, tested canned broad bean dips had significantly ($p < 0.001$) lower moisture content than that observed in fresh-made broad bean dips. Hence, canned broad bean dips had significantly higher carbohydrate content. The moisture content in canned food products including broad bean dips should be low to reduce water available for microbial growth, which in turn increases the shelf-life of the canned products [30]. A significant difference ($p < 0.001$) in titratable acidity and pH values was found in our study between fresh-made and canned broad bean dips, with the latter having lower values. This can be explained due to manufacturing practices used in the preparation of canned broad bean dips by adding other ingredients that reduce the pH values, hence increasing the shelf-life of these canned products [27, 28].

Chickpea and broad bean dips are considered good sources of vitamins, specifically water-soluble vitamins [32, 35]. Water-soluble vitamins consist of vitamin C and the B vitamin family: B1 (Thiamine), B2 (Riboflavin), B3 (Niacin), B5 (Pantothenic acid), B 6 (Pyridoxine), B7 (Biotin), B9 (Folate), and B12 (Cobalamin). The structure and water solubility characteristic of these vitamins makes them easily degradable and prone to leaching during various food processing such as canning [35]. It is noteworthy that vitamin B7 is widely spread in various foods, whereas vitamin B12 is mostly found in animal products. Furthermore, the retention of vitamin B9 and vitamin C in vegetable and fruit canned foods have been reported to be minimal, hence, the detection of these vitamins in canned foods is often considered not of nutritional concern [28, 36]. USDA Food Data Central Database (2019) reported that commercial *Hummus* (chickpea dips) contained B1 (0.16mg/100g), B2 (0.13mg/100g), B3 (1.02mg/100g), B5 (0.35mg/100g), and B6 (0.15mg/100g) [37]. In our study, fresh-made chickpea dips had higher contents of B5 (0.51mg/100g), while canned chickpea dips contained higher contents of B1 (0.46mg/100g) and B5 (0.87mg/100g) than those reported by the USDA Food Data Central database. Moreover, significant differences ($p < 0.001$) in all B vitamins tested were found between canned chickpea dips and fresh-made chickpea dips, with the former having higher

amounts. Studies have shown that various pretreatments of chickpeas in food processing improve or reduce their nutritional contents including the B vitamins [38, 39]. As for the broad bean dips, there are no reports by the USDA Food Data Central Database (2019) on these dips, however, the database showed that boiled broad beans (the main ingredient of broad bean dips) contained B1 (0.09mg/100g), B2 (0.09mg/100g), B3 (0.71mg/100g), B5 (0.16mg/100g), and B5 (0.07mg/100g) [33]. Canned broad bean dips in this study contained higher quantities of all tested B vitamins apart from B6 (0.04mg/100g) than that reported by the USDA Food-Data Central database. Moreover, significant differences ($p < 0.001$) in all B vitamins tested were found between canned broad bean dips and fresh-made broad bean dips, with the former having higher amounts except for B6. These differences may be due to the manufacturing processes such as fermentation and germination that improved the release of these vitamins in the canned products [38, 39].

## Limitations

The study's small sample size is an important limitation that should be acknowledged. This was due to the unwillingness of many commercial factories, restaurants, hospitals, and hotels found in Jordan to participate in this study. This could limit the study findings' generalizability. Future research should reconfirm our findings by conducting larger-scale studies.

## Conclusions

In conclusion, this study demonstrated that both fresh-made and canned chickpea and broad bean dips commercially made and sold in Jordan were excellent sources of proteins, carbohydrates, fibers, and B vitamins, and at the same time, these dips were low in fat. Significant differences were detected between fresh-made and canned chickpea dips, with the former showing higher nutrition values, titratable acidity, and tested B vitamins (except for B6). Similar results were also observed in canned broad bean dips. These detected differences may be due to many factors including the usage of various manufacturing processes. Further investigations are warranted to examine other nutrient differences between fresh-made and canned chickpea and broad bean dips, as well as the different manufacturing processes used in the preparation of these dips and their possible effects on the retention of these nutrients.

## Supporting information

**S1 Data.**
(XLSX)

## Acknowledgments

The authors wish to thank all the restaurants and the factories for their kind help and assistance in this study.

## Author Contributions

**Conceptualization:** Seham M. Abu Jadayil, Ali K. Alsaed.

**Data curation:** Seham M. Abu Jadayil, Ali K. Alsaed, Fatena Afaneh, Hanaa Khalaf.

**Formal analysis:** Ali K. Alsaed, Fatena Afaneh, Mohammed Z. Soudi.

**Funding acquisition:** Ali K. Alsaed.

**Investigation:** Seham M. Abu Jadayil, Ali K. Alsaed, Fatena Afaneh, Hanaa Khalaf.

**Methodology:** Fatena Afaneh, Hanaa Khalaf.

**Project administration:** Seham M. Abu Jadayil, Ali K. Alsaed.

**Resources:** Leena M. Ahmad, Mohammed Z. Soudi.

**Supervision:** Seham M. Abu Jadayil, Ali K. Alsaed.

**Validation:** Iman F. Mahmoud, Leena M. Ahmad.

**Writing – original draft:** Seham M. Abu Jadayil, Iman F. Mahmoud.

**Writing – review & editing:** Seham M. Abu Jadayil, Iman F. Mahmoud, Leena M. Ahmad.

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
