## [Decision Letter · Decision Letter 0]

26 Jun 2024

PONE-D-24-20137Proximate Analysis and B Vitamin Contents of Fresh-Made and Canned Chickpea and Broad Bean Dips Commercially Produced in JordanPLOS ONE

Dear Dr. Abu Jadayil,

Thank you for submitting your manuscript to PLOS ONE. After careful consideration, we feel that it has merit but does not fully meet PLOS ONE’s publication criteria as it currently stands. Therefore, we invite you to submit a revised version of the manuscript that addresses the points raised during the review process. Please submit your revised manuscript by Aug 10 2024 11:59PM. If you will need more time than this to complete your revisions, please reply to this message or contact the journal office at plosone@plos.org. Please include the following items when submitting your revised manuscript:A rebuttal letter that responds to each point raised by the academic editor and reviewer(s). You should upload this letter as a separate file labeled 'Response to Reviewers'.A marked-up copy of your manuscript that highlights changes made to the original version. You should upload this as a separate file labeled 'Revised Manuscript with Track Changes'.An unmarked version of your revised paper without tracked changes. You should upload this as a separate file labeled 'Manuscript'.

We look forward to receiving your revised manuscript.

Kind regards,

Md. Mahmudul Hasan

Academic Editor

PLOS ONE

Journal Requirements:

The Deanship of Research and Postgraduate Studies in Petra University 

We would like to express our deepest appreciation and gratitude to The Deanship of Research and Postgraduate Studies in Petra University for their continuous support. We also wish to thank all the restaurants and the factories for their kind help and assistant in this study. 

The Deanship of Research and Postgraduate Studies in Petra University 

5. We note that your Data Availability Statement is currently as follows: All relevant data are within the manuscript and its Supporting Information files.

Additional Editor Comments:

Please clarify the Table 2 & 4, 1 & 3. What is the different information there?Al tables contain all margin. Please only put header and footer of the margin.Please mention your innovations in the manuscript.What is the practical application of your research? Please mention in the manuscript.Why sample size is so small? How could these samples make conclusion of the quality of products?There are lots of brands in the market? How did you choose brands?How many batches tested?Writing is so poor. Need to improve.English is so poor. Need to improve.

Reviewers' comments:

Reviewer's Responses to Questions

**Comments to the Author**

1. Is the manuscript technically sound, and do the data support the conclusions?

Reviewer #1: Yes

2. Has the statistical analysis been performed appropriately and rigorously? 

Reviewer #1: Yes

3. Have the authors made all data underlying the findings in their manuscript fully available?

Reviewer #1: Yes

4. Is the manuscript presented in an intelligible fashion and written in standard English?

Reviewer #1: Yes

5. Review Comments to the Author

**Reviewer #1: **Dear author,

Thank you for the nice work. I have seen the detail of your work and my suggestion is please describe clearly the gaps why this finding is important and what gaps this finding will address. Discussion is shallow and please discuss in detail by using similar study conducted so far.

6. PLOS authors have the option to publish the peer review history of their article (what does this mean?). If published, this will include your full peer review and any attached files.

Reviewer #1: No

---

## [Author Response · Author response to Decision Letter 0]

6 Aug 2024

1. Thank you. Based on your comment, we have gone through the manuscript and ensured that it meets PLOS ONE's style requirements, including those for file naming. 

2. Based on your comment, we have added a statement regarding the permits we obtained for the work in the Materials and Methods Section. This statement was as follows: 

 “We communicated by email or phone with the most commercialized factories and restaurants specialized in chickpea and broad bean dips in Jordan (7 factories, 7 restaurants). Afterward, the research team visited several processing factories and restaurants producing chickpea and broad bean dips, as well as several hospitals and hotels. Only three processing factories and three restaurants agreed to participate in the study after explaining the study’s objectives, while the rest apologized.” Please see the track changes in Materials and Methods Section page 5, lines 94-102. 

3. Based on your comment regarding the funder’s role is as follows: 

4. We have removed the sentence” We would like to express our deepest appreciation and gratitude to The Deanship of Research and Postgraduate Studies in Petra University for their continuous support” from the Acknowledgments Section. Please see the track changes in Acknowledgments Section page 22, lines 437-439.

5. We have added a supplementary file containing data used in this study. 

Additional Editor Comments:

1. Tables 1 & 3 are the comparison between the three fresh samples alone and the three canned samples (chickpeas dips (Table 1) and broad bean dips (Table 3)). Whereas Tables 2 & 4 are comparison between fresh and canned (Table 1 for chickpea) and (Table 4 for broad bean dips).

2. All tables were revised and only headers and footers were added. Please see the track changes for Table 1 (Page 12), Table 2 (Page 13), Table 3 (Page 14), and Table 4 (Pages 15).

3. The innovation(s) of this study were added to the Discussion Section. Please see the track changes in Discussion Section page 17, lines 324-328.

4. The practical applications of this study were added to the Discussion Section. Please see the track changes in Discussion Section page 17, lines 321-327.

5. Many commercial factories, restaurants, hospitals, and hotels in Jordan which we approached for the study, declined participation, hence the sample size was small. 

We also added an extra section titled Limitation, to point out the limitations of this study including the sample size. Please see the track changes in Limitations Section page 22, lines 420-424.

6. The brands chosen for this study were based on: 1) Being the most commercialized brands produced and sold in Jordan, and 2) Being the products of the restaurants and factories which accepted to participate in this study.

7. Nearly 6 batches of chickpea dips and 6 batches of broad bean dips were selected randomly. Each batch contained 24 canned products. 

8. We have gone through the manuscript and made the improvements needed. 

Reviewer #1: 

Thank you. Based on your comment, amendments addressing the Discussion Section have been made. Please see the track changes in the Discussion Section pages 17-21, lines 315-418.

---

## [Editor Report · Decision Letter 1]

16 Sep 2024

Proximate analysis and vitamin B contents of fresh-made, canned chickpea and broad bean dips commercially produced in Jordan

PONE-D-24-20137R1

Dear Dr. Seham Abu Jadayil,

We’re pleased to inform you that your manuscript has been judged scientifically suitable for publication and will be formally accepted for publication once it meets all outstanding technical requirements.

Kind regards,

Md. Mahmudul Hasan, PhD

Academic Editor

PLOS ONE

Additional Editor Comments (optional):

The authors have addressed all the comments raised by the reviewers and revised accordingly. Therefore, the manuscript could be accepted for publication.
---

## [Editor Report · Acceptance letter]

19 Sep 2024

PONE-D-24-20137R1 

PLOS ONE

Dear Dr. Abu Jadayil, 

I'm pleased to inform you that your manuscript has been deemed suitable for publication in PLOS ONE. Congratulations! Your manuscript is now being handed over to our production team.

Kind regards, 

on behalf of

Dr. Md. Mahmudul Hasan 

Academic Editor

PLOS ONE